# Practical Considerations in the Management of Frail Older People with Diabetes

**DOI:** 10.3390/diseases13080249

**Published:** 2025-08-06

**Authors:** Dima Abdelhafiz, Ahmed Abdelhafiz

**Affiliations:** 1Department of Internal Medicine, Royal Shrewsbury Hospital, Mytton Oak Road, Shrewsbury SY3 8XQ, UK; dima.hafiz1@nhs.net; 2Department of Geriatric Medicine, Rotherham General Hospital, Moorgate Road, Rotherham S60 2UD, UK

**Keywords:** frailty, older people, diabetes mellitus, management, care plans

## Abstract

With increasing life expectancy, the number of older people living with comorbid diabetes and frailty is increasing. The development of frailty accelerates diabetes-related adverse outcomes. Frailty is a multidimensional syndrome with physical, mental and social aspects which is associated with increased risk of hypoglycaemia, dementia and hospitalisation. Therefore, regular screening for all aspects of frailty should be an integrated part of the care plans of older people with diabetes. In addition, every effort should be made for prevention, which includes adequate nutrition combined with regular resistance exercise training. In already frail older people with diabetes, metabolic targets should be relaxed and hypoglycaemic agents should be of low hypoglycaemic risk potential. Furthermore, the metabolic phenotype of frailty should be considered when choosing hypoglycaemic agents and determining targets. With increasing severity of frailty, proactive chronological plans of de-escalation, palliation and end-of-life care should be considered. These plans should be undertaken in a shared decision-making manner which involves patients and their families. This ensures that patients’ views, wishes and preferences are in the heart of these plans.

## 1. Introduction

The global prevalence of diabetes mellitus among adults aged 20–79 years is about 10.5% and is expected to increase by 46% by the year 2045 due to increased life expectancy [1]. About half of people with diabetes are 65 years of age or older and the prevalence is highest (24%) in those aged between 75 and 79 years [1]. In addition to traditional diabetes-related vascular disease, frailty emerges as a new complication affecting up to 32–48% of older people with diabetes [2,3]. Frailty is defined as a state of decreased physiological reserve at multiple organ levels that leads to a limited capacity to maintain homeostasis and increase vulnerability to physical or psychological stressors [4]. Diabetes is associated with increased risk, prevalence and burden of frailty [5,6,7]. The development of frailty in older people with diabetes leads to adverse outcomes such as higher risks of falls, injuries, fractures, dementia, disability, poor quality of life and mortality [8,9,10,11,12,13]. This leads to more demands of health care use such as hospitalisation and institutionalisation, which increases overall costs [14,15].

With the increasing ageing of the population, the prevalence of older people with diabetes and frailty is likely to increase. Frailty may interfere with the ability of older people with diabetes to self-care. There is little literature on the practical management of older people living with comorbid diabetes and frailty. Therefore, this manuscript reviews the relationship between frailty and diabetes, the impact of the emergence of frailty on diabetes management and practical considerations in the management of frail older people with diabetes.

### 1.1. Frailty–Diabetes Relationship

Diabetes-related complications and diabetes-associated morbidities play a crucial role in the development of frailty. Although multimorbidity overlaps with frailty, the multiorgan dysfunction concept leads to vulnerability of the individual to stressors and progression to frailty. In comparison to morbidity, frailty is associated with general weakness [16]. However, both conditions follow the sequence of disease, impairment, functional decline and disability characteristic of the disablement process model (DPM) pathway [17]. Multiple morbidities synergistically mediate the progression to frailty. For example, the association of diabetes with frailty was incremental when renal impairment, hypertension or any diabetic complication was also present [18,19]. Geriatric syndromes such as chronic pain, physical dysfunction, falls, cognitive impairment, depression and urinary incontinence are common in older people with diabetes and are associated with increased risk of frailty [20]. Other factors such as poor nutrition, inadequate protein intake and reduced physical exercise may contribute to muscle weakness and development of frailty [21]. Diabetes may directly lead to frailty through its negative effect on skeletal muscle mass and muscle quality and its accelerated ageing process [22,23]. In addition, fluctuations in blood glucose levels can lead to the development of frailty. For example, the risk of frailty increases proportionally with increased blood glucose levels or repeated episodes of hypoglycaemia [24]. In a reciprocal relationship, diabetes prevalence is higher in frail compared with non-frail subjects [25]. Therefore, diabetes and frailty are interrelated, and factors such as chronic low-grade inflammation and cardiovascular risk factors could constitute common pathways for both conditions [26,27].

### 1.2. Frailty and Adverse Outcomes

The development of frailty accelerates the occurrence of adverse outcomes in frail compared with non-frail older people with diabetes. Data from a large UK Biobank (20,566 participants) study showed an association between frailty measures and major adverse cardiovascular events (MACEs), hypoglycaemia, falls or fractures and mortality [28]. Frailty increased the risk of vascular events, all-cause and cardiovascular mortality and hypoglycaemia in the ADVANCE study. In addition, it attenuated the benefits from blood pressure lowering and intensive glycaemic therapy [29]. In the Look AHEAD study, the increase in frailty incidence was associated with poor cognitive function, physical function and mortality after 8 years of follow-up [30]. Frailty increased the risk of emergency department visits, hospitalisation, institutionalisation, fractures and mortality in older people with diabetes [31]. Frailty also accelerated the progression to end-stage renal disease in patients with diabetic kidney disease in a dose–response manner [32]. In a very old (≥80 years) cohort of people with diabetes and acute coronary syndrome, frailty was associated with increased incidence of death or 6-month hospital readmission [33]. Pre-frailty and frailty were associated with increased cardiovascular events, increased emergency department visits, health care utilisation, hospitalisation and mortality [34,35]. The progression of frailty can lead to disability. This could be potentiated by the development of cognitive impairment, and both conditions increase the risk of mortality [36] (Figure 1).

### 1.3. Frailty and Hypoglycaemia

Frailty is generally perceived as a shrinkage syndrome that is associated with weight loss and malnutrition. Malnutrition markers such as low albumin, low cholesterol, low haemoglobin and limb circumference were associated with frailty [37]. Anorexia of ageing, which is prevalent in about 30% of older people in communities and 85% of residents in care homes, is linked to both frailty and malnutrition [38]. Weight loss and malnutrition in frail older people with diabetes increase the risk of hypoglycaemia [39]. Therefore, the increased mortality due to low glycaemia demonstrated in randomised clinical trials was likely related to the underlying frailty and malnutrition, rather than to hypoglycaemia [40,41]. Therefore, frailty appears to be a confounding factor in studies relating mortality directly to hypoglycaemia.

This was confirmed in studies that showed an increased risk of mortality in older people with low HbA1c. The frail participants in these studies had evidence of inflammation, such as elevated ferritin levels, and evidence of malnutrition, such as low cholesterol and low serum albumin, which suggest frailty-associated low glycaemia [42]. Therefore, in certain older people, especially those with recurrent hypoglycaemia, it is appropriate to consider low HbA1c as a biochemical marker of frailty and a surrogate marker of “burnt out diabetes”, rather than a direct cause of adverse outcomes [43].

### 1.4. Frailty and Dementia

Unhealthy lifestyle, oxidative stress and impaired repair are common factors that link frailty to dementia [44,45]. Increasing age and the development of dementia accelerate the progression of frailty [46]. Diabetes increases the risk of dementia directly due to metabolic abnormalities, likely due to oxidative stress and chronic inflammation. In addition, diabetes indirectly increases the risk of dementia due to the increased prevalence of diabetes-related cardiovascular complications [47,48]. The presence of frailty is a predictor of dementia [10]. The simultaneous occurrence of frailty and cognitive impairment has been described as ‘cognitive frailty’ by the International Association of Gerontology and Geriatrics (IAGG) [49]. This terminology may help early diagnosis and highlights that both conditions could be parts of a wider spectrum [50]. It has been shown that the co-existence of frailty and cognitive impairment increases adverse outcomes such as mortality in a synergistic manner [36]. Hypoglycaemia may be linked to both conditions [51]. Hypoglycaemia is associated with reduced physical activity and increased anxiety, fear, stress and depression that lead to social isolation, worsening frailty and cognitive function [52,53]. Therefore, hypoglycaemia appears to be a risk factor of frailty in its three dimensions of physical, mental and social aspects.

## 2. Practical Considerations

### 2.1. Frailty Assessment

Frailty is an expression of biological rather than chronological age and is not a synonymous or inevitable consequence of ageing [54]. It is a multidimensional condition that includes physical, psychological, biological, economic and social domains [55]. With increasing life expectancy and increased prevalence of diabetes, frailty’s prevalence is set to increase. Therefore, comprehensive geriatric assessment (CGA) that includes assessment of the multidimensional aspects of frailty is required in everyday clinical practice. The frailty assessment tools that are validated in older people with diabetes are summarised in Table 1 [56,57,58,59,60,61,62,63]. These tools have variable predictive ability to associate frailty with outcomes such as glycaemic control, hypoglycaemia, disability and mortality in older people with diabetes. Despite the diversity of frailty assessment tools, there is no recommended standard assessment tool for older people with diabetes [64]. The three most commonly used screening tools for identifying frailty in older people with diabetes, associated with the most outcomes, are the Fried physical frailty phenotype, the clinical frailty scale and the frailty index [65]. The five criteria of the Fried phenotype include physical decline, loss of weight, weakness, slowness and exhaustion [56]. The nine points of the clinical frailty scale are descriptions of functional abilities, which predict mortality [57]. The frailty index uses clinical criteria derived from the comprehensive geriatric assessment [58] (Table 1). In addition to frailty assessment, patients should be screened for their physical, mental and social functions on a regular basis. Examples of screening tools for these functions are summarised in Box 1 [66,67,68,69,70,71].

Box 1Screening for physical, mental and social functions in frail older people with diabetes mellitus [66,67,68,69,70,71].
**A. Physical function**Barthel Index to assess patient ability in performing activities of daily living by assessment of 10 different tasks; score ranges between 0 and 100:**Score:** ●91–100 independent; ●61–90 moderate dependence; ●21–60 severe dependence; ●0–20 total dependence.**B. Mental function** ***1. Cognition (Mini-Cog)***Ask the patients to repeat three items such as key, lemon and balloon, then provide a clock face and ask them to  ●Draw the numbers of the clock face; ●Draw the hands of the clock to show time as ten to three; ●Recall the three items.**Score:** One mark for each task performed and for each item remembered; a score ≤3/5 defines cognitive impairment. ***2. Depression (PHQ-2)***Ask the patients whether they have ●Little interest in doing things? ●Been feeling down, depressed or hopeless?**Score:** Any positive answer triggers assessment using the detailed (PHQ-9) ***3. Anxiety (GAD-2)***Ask the patients whether, over last 2 weeks, they have been ●Feeling nervous, anxious or on edge? ●Unable to stop or control worrying?**Score:** Give 0 for not at all, 1 for several days, 2 for >half the days, 3 for nearly every day; a score of ≥3/6 defines anxiety. ***4. Distress (PAID-1)*** ●Is the patient is worrying about the future and the possibility of serious complications?**Score:** A positive answer suggests underlying diabetes-related emotional distress.**C. Social function**Social Support Rating Scale (SSRS): A 10-item questionnaire to assess individuals’ social support. It covers three dimensions: objective support, subjective support and support utilisation.Higher scores indicate greater social support, while lower scores indicate a need for interventions to improve support or address potential social isolation.Subscale scores (objective, subjective and utilisation) identify which aspects of social support might be lacking or need strengthening.



### 2.2. Lifestyle

Lifestyle intervention can improve physical function and frailty index scores, and reduce metabolic risk factors, polypharmacy and health-care-related costs, as demonstrated in the Look AHEAD study [72]. Similarly, nutrition and exercise intervention improve cognitive function, as reported by the FINGER study [73].

The nutritional needs of frail older people may vary and may be striking in people with a loss of the ability to prepare a meal, loss of appetite or those with poor oral health [74]. These deficits may accelerate frailty-related problems such as sarcopenia [75]. Therefore, diet in frail older people should be nutritionally adequate and balanced in micro- and macronutrients to preserve lean body mass [76]. A fruit- and vegetable-rich diet reduced frailty risk in women [77]. Due to their catabolic state and anabolic resistance, older people require higher intake of proteins, especially branched-chain essential amino acids such as leucine, to help preserve their muscle mass and avoid sarcopenia [78]. Leucine, combined with vitamin D, can improve muscle mass and function [79]. Daily protein intake should be 1.0–1.2 g/kg and up to 2.0 g/kg in severely frail people with malnutrition [80]. In addition to adequate nutrition, progressive resistance training (PRT) exercise further improves muscle profile and function [81]. The combination of PRT with adequate nutrition increases muscle performance [82]. The NuAge study demonstrated the superiority of combined good-quality diet and exercise [83].

Specialised educational programmes about smoking cessation, reduction in alcohol consumption and diabetes self-care may improve metabolic profile and reduce functional decline in older people compared to usual care [84].

### 2.3. Glycaemic Control

There are no clinical trials or substantial evidence to suggest what level of glycaemic control is best in frail older people with diabetes. Most of the available studies are either cross-sectional or retrospective, describing association rather than causation in glycaemia and frailty. One method of analysis is to look at what is the best glycaemic level to reduce the development of frailty in non-frail older people with diabetes. The other way is to look at what is the most suitable glycaemic control in already frail older people with diabetes. Several studies explored the effects of glycaemic control on the development of frailty [85,86,87,88,89,90,91,92,93,94,95]. Some studies showed that the lower the blood glucose and HbA1c levels, the more severe the frailty [87,88,92]. The studies suggesting an increased risk of frailty with low glycaemia have a few limitations. For example, dementia, which could be a confounding factor, was the driving factor for care need in patients with low HbA1c in one study [87]. Another study demonstrated not only low HbA1c to be associated with frailty, but also low body weight, low serum albumin, low Hb and low cholesterol, suggesting that malnutrition could be the underlying factor in this relationship [88]. The study reporting moderate or severe frailty associated with low blood glucose included only patients with acute admission. Their low blood glucose could be due to underlying undernutrition or due to the fact that acute illness such as sepsis and hepatic or renal failure may reduce blood glucose levels. In addition, participants were recruited from a single unit rather than the whole hospital, which may limit generalisability [92].

Other studies showed uncontrolled diabetes, defined as persistent hyperglycaemia or HbA1c ≥ 8.0%, to be associated with frailty [93,94,95]. The strengths of the first study are its prospective design, the use of continuous glucose monitoring (CGM) and the detailed documentation of fluctuations in blood glucose levels [93]. The other two studies are limited by their cross-sectional design and small number of frail participants [94,95]. Studies have also demonstrated a proportional relationship between high blood glucose and the risk of frailty in a dose–response manner [85,89,90,91]. The strengths of one study are its prospective design and the objective assessment of diabetes and frailty according to standard criteria [85]. The strengths of the other study are its prospective design and adjustment for multiple confounders, although the association of HbA1c with frailty was significantly attenuated after adjustment for confounders. This may suggest that confounding factors may affect the relationship of uncontrolled diabetes and frailty [89].

One study demonstrated a U-shaped relationship between HbA1c and frailty, with levels lower or higher than 7.6% associated with increased risk of frailty, suggesting that frailty could be associated with extreme levels of blood glucose in both directions [86]. The limitations of this study are its mixed ethnic makeup, self-reporting and irregular intervals of blood glucose level measurement. Most importantly, the study did not report the morphological phenotypes of the two groups of participants. It is plausible that the population with high HbA1c are on the overweight or obese side and those with low HbA1c are on the underweight side of frailty phenotypes. Therefore, the presence of different metabolic phenotypes could explain the apparent contradiction that both high and low HbA1c increase the risk of frailty.

The participants in the studies associating low glycaemia with frailty have a high prevalence of malnutrition, which could be an explanation of this relationship [87,88]. Studies reporting poor glycaemic control and time spent in hyperglycaemia accelerating the risk of frailty in non-frail obese subjects could be explained by increased chronic inflammation and increased insulin resistance [85,90]. Therefore, for reducing the risk of frailty in non-frail subjects, good glycaemic control of HbA1c around 7.5% and reduction in blood glucose level variability is required. For already frail older people, a target that ranges from 7.5 to 8.5% is reasonable depending on function [96]. (Figure 2) This population is unlikely to benefit from tight glycaemic control due to the competing effects of their associated morbidities [97]. As the severity of frailty increases and function declines, targets should focus on short-term day-to-day blood glucose levels between 6 and 15 mmol/L rather than a long-term HbA1c target, as values outside this range may result in cognitive changes [98] (Table 2).

### 2.4. Choice of Hypoglycaemic Therapy

Similarly to glycaemic control, the choice of hypoglycaemic therapy can be viewed from the point of view of medication which can delay the development of frailty in non-frail patients and agents and which is suitable in already frail older people. Metformin is the most studied drug, which shows better muscle mass index, faster gait speed, improved handgrip strength, low risk of frailty and slowed down the development of age-related morbidities such as cardiovascular disease, cancer, dementia, depression and injurious falls [99,100,101]. In one study, Thiazolidinedione (TZD) users showed attenuated decline in the loss of walking speed compared with non-users [102]. The effects of sulfonylureas (SUs) on muscle function are inferior to metformin [103]. Meglitinides and the SU glibenclamide may be associated with muscle atrophy compared to metformin [104,105]. Dipeptidyl peptidase-4 (DPP-4) inhibitors may have a beneficial effect on muscle function [106]. The effects of sodium glucose cotransporter-2 (SGLT-2) inhibitors and glucagon like-1 receptor agonists (GLP-1RAs) is not clear. SGLT-2 inhibitors and GLP-1RAs significantly reduce body weight and, as a result, may reduce muscle mass. Some SGLT-2 inhibitors have shown a reduction in skeletal muscle mass while others have not, but they may improve muscle quality rather than muscle mass, which may have an overall positive effect [107,108,109,110,111]. A meta-analysis of 18 randomised trials of SGLT-2 inhibitors reported an overall significant reduction in skeletal muscle mass (−1.01 kg, 95% CI −1.91 to −0.11). However, the studies included in the meta-analysis were of short duration (median 24 months) and most of them had small sample sizes [112]. There is emerging evidence to suggest that SGLT-2-associated weight reduction is not associated with muscle mass or muscle strength loss. The EMPA-ELDERY randomised controlled study included a total of 129 older people with diabetes, with a mean (SD) age of 74.1 (5.0). After 52 weeks of follow-up, the placebo-adjusted difference was −2.37 kg (95% CI −3.07 to −1.68) in body weight and −1.84 kg (−2.65 to −1.04) in fat mass in the empagliflozin group compared to the placebo group. However, the placebo-adjusted change was not significant in terms of muscle mass −0.61 kg (95% CI −1.61 to 0.39) or grip strength −0.3 kg (95% CI −1.1 to 0.5) [113]. An increased prevalence of falls was shown in users of GLP-1RAs compared to other agents, but their effect on frailty was not clear [114]. In a meta-analysis, GLP-1 RA reduced body weight, fat mass and lean mass. The absolute lean mass loss was 25% of the total weight loss, but the relative lean mass, defined as the percentage change from baseline, was the same. Among GLP-1RAs, liraglutide was the only one to achieve significant weight loss without significantly reducing lean mass. Tirzepatide and semaglutide were the most effective at reducing weight and fat but the least effective at preserving muscle mass [115]. In animal studies, GLP-1RAs have shown positive effects on metabolism by reducing insulin resistance and improving β-cell function [116,117]. Whether this favourable metabolic effect will lead to better lean muscle mass and muscle function will need further exploration. In addition to the cardio-renal protective effects of SGLT-2 inhibitors and GLP-1RAs, they also appear to have pleotropic properties that may improve the overall clinical functions of frail older people with diabetes beyond glycaemic control. Empagliflozin has been shown to improve cognitive (Montreal Cognitive Assessment score) and physical (5 m gait speed) function in frail older people with diabetes as soon as three months after treatment begins. This is likely due to its attenuation of mitochondrial oxidative stress in human endothelial cells [118].

Although there is no direct data on its relation to frailty, insulin, through its anabolic properties, may have some positive effects on skeletal muscle function [119].

For the choice of hypoglycaemic therapy in frail older people, safety comes as a priority for glycaemic targets. Metformin remains the first-line therapy due to its safety profile, low risk of hypoglycaemia and other multiple benefits independent of glycaemic control [99,100,101]. DPP-4 inhibitors have a low risk of hypoglycaemia and good safety profile, but there is a marginal non-significant increase in heart failure risk with saxagliptin [120]. TZDs, although they have a low risk of hypoglycaemia, should be used with caution in patients with heart failure due to the increased risk of fluid retention [121]. Meglitinides and SUs are better avoided in frail older people due to the high risk of hypoglycaemia. The use of SGLT-2 inhibitors and GLP-1RAs is likely affected by the underlying frailty metabolic phenotype. Frailty is a metabolically heterogeneous condition that spans across a spectrum with two distinctive metabolic phenotypes at both ends. A sarcopenic obese (SO) phenotype, characterised by unfavourable metabolism and increased insulin resistance, at one end and an anorexic malnourished (AM) phenotype, characterised by significant weight loss and reduced insulin resistance, at the other end [122]. SGLT-2 inhibitors and GLP-1RAs are safe, with minimal risk of hypoglycaemia, and showed significant cardio-renal benefits in the SO frailty phenotype due to its high baseline cardiovascular risk. However, they are better avoided in the AM frailty phenotype due to the high risk of weight loss, dehydration and hypotension [123]. In the AM phenotype, the early use of long-acting insulin analogues can be considered due to their anabolic properties, low risk of hypoglycaemia and the convenience of once-daily injections [119]. The advantages and disadvantages of hypoglycaemic agents in frail older people with diabetes are summarised in Table 3.

### 2.5. Reducing Hypoglycaemia

Hypoglycaemia is less symptomatic in older people with diabetes, and when symptoms develop, they are generally non-specific. Symptoms may be atypical and may be attributed to old age rather than to hypoglycaemia [124]. Another diagnostic challenge is the presentation of hypoglycaemia with agitation, increased confusion or behavioural changes that may be misdiagnosed as dementia. Other symptoms of hypoglycaemia such as nausea, falls or unsteadiness were reported in primary care in older people with diabetes treated with insulin who had a previous history of hypoglycaemia [125]. These diagnostic difficulties may lead to the under-reporting of hypoglycaemia [126]. Therefore, the clinical diagnosis of hypoglycaemia in this age group may be difficult. Educational programmes for patients, carers and health care professionals should be in place to facilitate the early recognition of atypical presentations of hypoglycaemia. The risk of hypoglycaemia is likely to increase in the AM, rather than the SO, frailty phenotype due to the associated anorexia and weight loss. In addition, this phenotype is associated with a high prevalence of dementia that is associated with erratic eating patterns, which further increases the risk of hypoglycaemia [127]. Short-acting insulin, given before meals, should be considered in patients who have erratic eating habits to reduce hypoglycaemia risk. Agents with high hypoglycaemia risk should be reviewed and stopped. For example, SUs have been shown to be inappropriately used in very old (≥80 years) patients with multiple morbidities, including cognitive impairment, who have low HbA1c (5.9%) and a history of recurrent hypoglycaemic episodes [128].

Glycaemic targets should be relaxed and intensive treatment should be avoided. In a retrospective analysis, intensive treatment was prevalent in 20% of patients and increased incidence of severe hypoglycaemia was observed in clinically complex patients (defined as ≥75 years old with dementia, end-stage renal disease or ≥3 comorbidities) [129]. Despite these relaxed targets, CGM showed that asymptomatic hypoglycaemia can still occur in older people with diabetes regardless of HbA1c levels, suggesting that hypoglycaemia is not necessarily related to HbA1c [130]. Therefore, CGM is useful in older people treated with insulin but may have less benefit in patients on oral hypoglycaemic agents [131]. Annual reviews of frail older people with diabetes should include identification of risk factors of hypoglycaemia such as polypharmacy, impaired organ function, multiple comorbidities, especially cognitive dysfunction, and SU or insulin therapy. Other social factors that may predispose a person to hypoglycaemia, such as isolation and a lack of carers, should be looked at on regular bases. Every effort should be made to simplify medication regimens and to use the therapy with the lowest hypoglycaemia risk (Box 2).

Box 2Reducing hypoglycaemia.

**
H
**
**
ypoglycaemia risk assessment
**

**
Y
**
**
early review of risk factors
**

**
P
**
**
olypharmacy reduction
**

**
O
**
**
ptimisation of therapy 
**

**
G
**
**
oals of therapy setting 
**

**
L
**
**
ax glycaemic targets
**

**
Y
**
**
earlong adherence to lifestyle and meal time compliance
**

**
C
**
**
ontinuous glucose monitoring in appropriate patients
**

**
A
**
**
void drugs with high hypoglycaemic potential 
**

**
E
**
**
ducation of patients and carers
**

**
M
**
**
onitoring of organ function
**

**
I
**
**
nsulin regimen simplification
**



### 2.6. Avoiding Hospitalisation

Risk factors that could lead to the hospitalisation of frail older people with diabetes should be addressed and avoided. For example, with increasing severity of frailty and weight loss, the need for medications declines, and this should be reviewed regularly to avoid overtreatment that leads to side effects and hospitalisation. Other medications such as renin–angiotensin–aldosterone system (RAAS) blockers can lead to acute kidney injury (AKI), hyperkalaemia or hypotension, with further deterioration in renal function, especially in those with chronic kidney disease (CKD). Withdrawal of these agents has been shown to improve kidney function [132]. The use of nonsteroidal anti-inflammatory drugs (NSAIDs) should be with caution in frail older people with diabetes and CKD due to the increased risk of AKI. In addition, patients requiring imaging with radio contrast agents should be well hydrated before the procedure and their kidney function should be carefully monitored thereafter. In general, hydration should be maintained in frail older people with diabetes due to impaired thirst sensation in this group in order to avoid the risk of volume depletion and hyperglycaemic crises. Polypharmacy is common as people get older, which may lead to an increased risk of drug errors, hypoglycaemia and hospitalisation. Therefore, regular medication review should be considered to reduce costs, minimise adverse drug reactions and reduce hospitalisation [133]. As frailty is a risk factor for falls and hospital admission, CGA is important to screen for fall risk and enable the implementation of exercise programmes to reduce fall-related hospitalisation [134]. The addition of CGA to the annual review of frail older people with diabetes will help screen for the presence of depression or cognitive impairment, which may lead to difficulties in self-care and poor glycaemic control that may result in hospitalisation [135,136]. In addition, CGA will pick up any issues in social aspects of frailty in order to avoid hospitalisation related to a break down in social care. As frailty is associated with reduced immunity, annual vaccination, along with good glycaemic control, may help reduce infection-related hospitalisation [137]. When hospitalisation occurs, discharge documents should include a problem list and a management plan that is communicated appropriately to the community care team to reduce risk of readmission [138]. A community diabetes nurse case manager or long-term telephone support could be very efficient in the general follow-up of patients to improve quality of life and reduce hospitalisation [139,140]. For people residing in care homes, the introduction of advanced nurse practitioners may help improve staff confidence and reduce hospital admissions [141] (Box 3).

Box 3Avoiding hospitalisation. CGA = Comprehensive geriatric assessment, RAAS = Renin–angiotensin–aldosterone system, NSAIDs = nonsteroidal anti-inflammatory drugs, CKD = Chronic kidney disease.

**
H
**
**
ypoglycaemic medication review
**

**
O
**
**
n discharge, community diabetes-specific service follow-up
**

**
S
**
**
creen for depression, dementia and fall risk
**

**
P
**
**
olypharmacy reduction
**

**
I
**
**
nsulin administration ability regular review
**

**
T
**
**
ailored and individualised care plans
**

**
A
**
**
nnual CGA review
**

**
L
**
**
iaison with diabetes specialist nurses
**

**
I
**
**
nput from dieticians to reduce malnutrition and maintain hydration
**

**
S
**
**
pecific diabetes-related training for care home staff
**

**
A
**
**
void RAAS inhibitors and NSAIDs in CKD patients
**

**
T
**
**
raining and education for patients and carers
**

**
I
**
**
ntegrated primary, secondary and care home services
**

**
O
**
**
ptimisation of exercise programmes to improve mobility and balance
**

**
N
**
**
eed for vaccination in high-risk patients
**



### 2.7. Care Needs

Frail older people with diabetes will require extra care considering their complex morbidity burden and functional decline, which may affect their ability to perform independent self-care tasks. Most importantly, routine checks by carers may help to identify early signs of deterioration or complications such as hypoglycaemic episodes. Care can be formal, delivered by paid professionals, or informal, voluntarily offered by family members, neighbours or friends. Care can be delivered directly or remotely through the monitoring of patient compliance and outcomes. Carers’ roles can range from simple emotional support, such as providing company or help in daily activities, to diabetes-specific medical care, such as the administration of medications including insulin injections and regular blood glucose monitoring. Carers should be educated about the lifestyle needs of frail older people with diabetes and their daily routine tasks. They should be able to recognise acute complications of diabetes and have easy and quick access to specialised care if urgently required. Carers’ involvement will have a positive effect on outcomes such as improvement in patients’ self-efficacy, perceived social support, diabetes knowledge and diabetes self-care [142]. Remote monitoring care can also result in a positive impact on clinical, economic and humanistic outcomes ranging from diabetes-related parameters to patient compliance [143].

In addition to care for patients, it is important to care for carers, as delivering care may affect their health physically, socially and psychologically. Therefore, it is vital to check carers’ burden and its effect on their own health. This will help identify their needs and their ability to continue providing the required level of care [144]. Carers’ burden can also put carers at risk of worsening their own medical conditions, developing new ones or leading to significant morbidity and risk of mortality. In addition, this may lead to the deterioration of patients’ function, disability and institutionalisation [145]. For example, in a study including frail older people with diabetes with a mean (SD) age of 83.1 (4.9) years, 81.4% of carers reported severe overburden [146]. Another study reported that reducing carers’ burden will have better health outcomes [147]. Carer support is a key consideration from the outset of initiating care for frail older people with diabetes. It should be available and provided whenever needed and escalated in parallel with the progression of patients’ conditions. Support plans, which set out the carers’ needs, wishes around providing care and balance with their own life, should be developed in partnership with the carers. These should also be regularly reviewed to reduce the burden on their own health [148].

### 2.8. De-Intensification

With the development of frailty and accumulation of comorbidities, the risk of overtreatment and hypoglycaemia increases, especially in the AM frailty phenotype with significant weight loss. The significant weight loss in this frailty phenotype may lead to the resolution of hyperglycaemia and normalisation of HbA1c, a condition termed “burnt out diabetes” [149]. Therefore, in this frailty phenotype, health care professionals should focus on stepping down medications and using simple regimens of therapy. A patient-based approach is the key for the success of de-intensification to maintain comfort and quality of life. Triggers for de-intensification are unintentional weight loss, tight glycaemic control (HbA1c < 7.0%) or recurrent episodes of hypoglycaemia. De-intensification should be performed gradually and regularly re-checked for the need to restart therapy if required. In certain patients, complete withdrawal of therapy may be considered. Hypoglycaemic medications have been safely withdrawn in frail patients with type 2 diabetes without deterioration of their glycaemic control [150,151]. The main characteristics of these patients were significant weight loss, multiple morbidities, polypharmacy and recurrent hypoglycaemia. In AM frail patients who are not tolerant to oral hypoglycaemic therapy due to erratic eating patterns, a single daily dose of a long-acting insulin analogue can be an alternative, which may help control symptoms without inducing hypoglycaemia if carefully titrated. In the SO frailty phenotype with high cardiovascular risk, intensification rather than de-intensification of therapy is required, especially the use of SGLT-2 inhibitors and GLP-1RAs due to their cardio-renal benefit, regardless of HbA1c levels (Figure 3).

Statins can be discontinued in very frail AM patients who have low serum cholesterol and low albumin levels due to malnutrition, low atherosclerotic cardiovascular disease (ASCVD) risk and short life expectancy [152]. Discontinuation can also be considered in patients ≥ 85 years old who were prescribed a statin for primary prevention. Statins should be continued at their full current or lower dose in SO frail patients with high ASCVD burden [153]. Blood pressure (BP) medication de-intensification is possible, and a BP target of 150/90 mmHg is accepted in patients > 80 years old [154]. In this age group, higher systolic BP may be associated with lower risk of cognitive decline and mortality [155,156]. Aspirin can be continued, as a secondary prevention, in frail older people with diabetes and high ASCVD risk if tolerated with no major bleeding risk. However, it can be withdrawn in AM frail phenotype patients who are at risk of bleeding and have short life expectancy [157]. For primary prevention, aspirin use can be withdrawn in people ≥ 70 years old after discussion of the risks and benefits [158].

### 2.9. Palliation and End of Life

With increasing age and severity of frailty, health care professionals should move from target achievement practice to a symptom control approach, which put symptoms before targets. Although the recognition of the palliative phase in frail older people with diabetes may not be straightforward, health care professionals should not overestimate survival [159]. The aim of glucose-lowering therapy in this phase of life should focus on reducing variability in blood glucose levels to avoid development of symptoms, which may lead to decompensations and unnecessary emergency hospital visits. Patients’ conditions should be regularly monitored, and the timing of palliation should be considered without delay [160]. The decision of palliation should be taken in the context of CGA by a multidisciplinary team with the clear involvement of patients and families in a shared decision-making manner. The aim of palliation is to reduce symptoms and medication burden, avoid unnecessary invasive interventions, preserve function and maintain a good quality of life. For patients in the palliative phase, proactive discussion and documentation of advanced care planning and wishes for end-of-life care are important [161]. The aims in end-of-life care are to maintain comfort, reduce suffering and provide a setting of care in accordance with patients’ wishes.

To achieve comfort, no dietary restrictions should be in place, and hypoglycaemic therapy can then be adjusted to patients’ choices of food. Invasive therapy such as sliding-scale insulin should be avoided, and the insulin regimen should be simplified. For example, short-acting insulin can be given as required only after, not before, meal consumption. When patients reach the terminal phase of life with unreliable oral intake, oral hypoglycaemic therapy can be withdrawn in patients with type 2 diabetes and once-daily long-acting basal insulin can be given instead. In patients with type 1 diabetes, the insulin dose can be reduced and given only after food consumption, with the continuation of long-acting insulin. When life expectancy is only days, insulin can be stopped in type 2 diabetes but continued with a further reduction in dose in type 1 diabetes. When life expectancy is only hours and patients are unconscious, insulin can be stopped in type 1 diabetes as well (Figure 4).

## 3. Conclusions

Frailty is an increasingly recognised complication of diabetes. With increasing life expectancy, the number of older people living with frailty and diabetes is likely to increase. Frailty increases diabetes-related adverse outcomes and is associated with an increased risk of hypoglycaemia, dementia and health care resource utilisation, including hospitalisation. Therefore, routine screening for frailty and preventative programmes should be a routine part for care of older people with diabetes to reduce the risk of frailty. For frail older people, relaxed metabolic targets should be in place, with appropriate choices of medications which have less side effects, especially hypoglycaemia. Care should be individualised and tailored to patients’ needs. As frailty progresses, timely de-intensification, palliation and end-of-life plans should be proactively put in place. These care plans should be jointly discussed with patients and their families in a shared decision-making manner, incorporating their wishes and preferences.

### 3.1. Future Perspectives

Diabetes itself, diabetes-related complications and diabetes-associated comorbidities are risk factors for the development of frailty, which subsequently leads to disability and mortality. Therefore, future research to explore methods of frailty prevention is important, and this should include investigating what glycaemic level is best for frailty prevention. The extra glycaemic benefits of GLP-1RAs and SGLT-2 inhibitors need to be investigated in future clinical trials where frailty is the main outcome. In addition, novel hypoglycaemic agents with effects on muscle function are required in order to reduce muscle mass loss. For example, myostatin antibodies may increase muscle mass, decrease body fat and improve physical function in older people with sarcopenia, a condition closely linked to frailty [162]. Other experimental hormonal therapies which need further investigation are growth hormones, growth hormone releasing hormone analogues, androgen receptor agonists in men and hormone replacement in post-menopausal women [163,164]. An important point is the metabolic heterogeneity of frailty, which will affect the choice of hypoglycaemic therapy as well as the decision about the de-intensification of therapy. There is little literature on the de-intensification of therapy in terms of patient characteristics and the timing of de-intensification. It appears that SO frail patients will continue to benefit from therapies such as GLP-1 RAs and SGLT-2 inhibitors, while AM frail patients should be considered for de-intensification. Therefore, future studies should include the characteristics and metabolic profiles of frail older people with diabetes from the outset of the trials. Furthermore, a full exploration of the metabolic spectrum and sub-stratification of frailty still needs future investigation.

### 3.2. Key Points

With increasing life expectancy, the number of older people living with diabetes and frailty is likely to increase.Frailty will increase diabetes-related complications, especially hypoglycaemia, dementia and hospitalisation.Regular screening for frailty should be a routine part of the care plans of older people with diabetes.Relaxed metabolic targets and the avoidance of agents with high hypoglycaemic potential should be considered in frail older people with diabetes.With increasing severity of frailty, de-intensification, palliation and end-of-life plans should be in place after discussion with patients and their families.

## Figures and Tables

**Figure 1 diseases-13-00249-f001:**
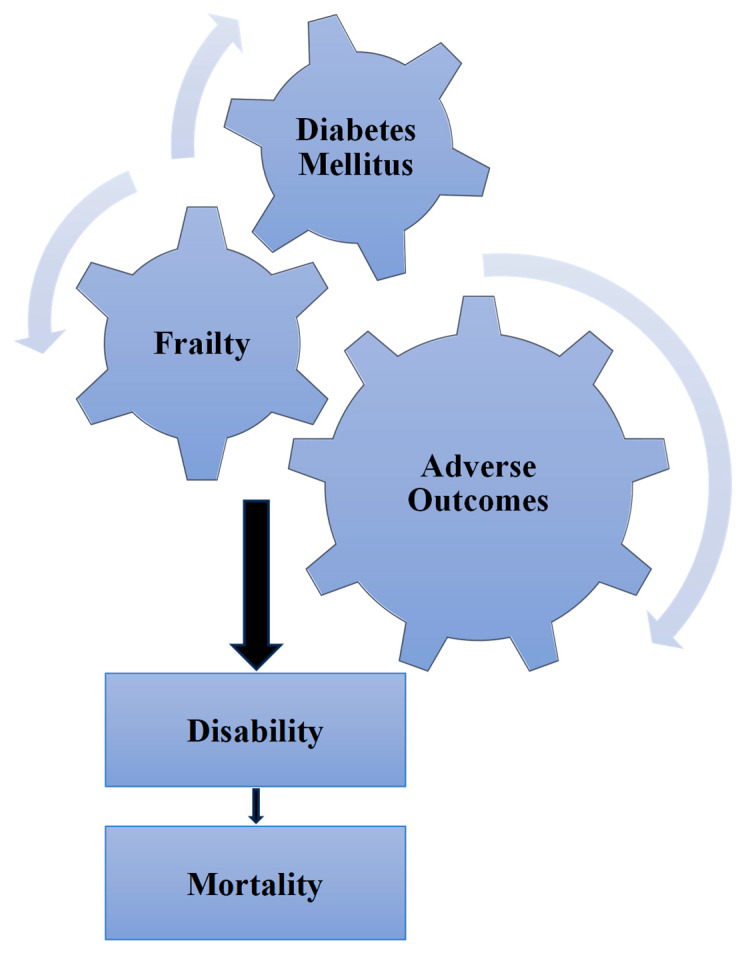
Frailty accelerates diabetes-related adverse outcomes, leading to increased risk of disability and mortality.

**Figure 2 diseases-13-00249-f002:**
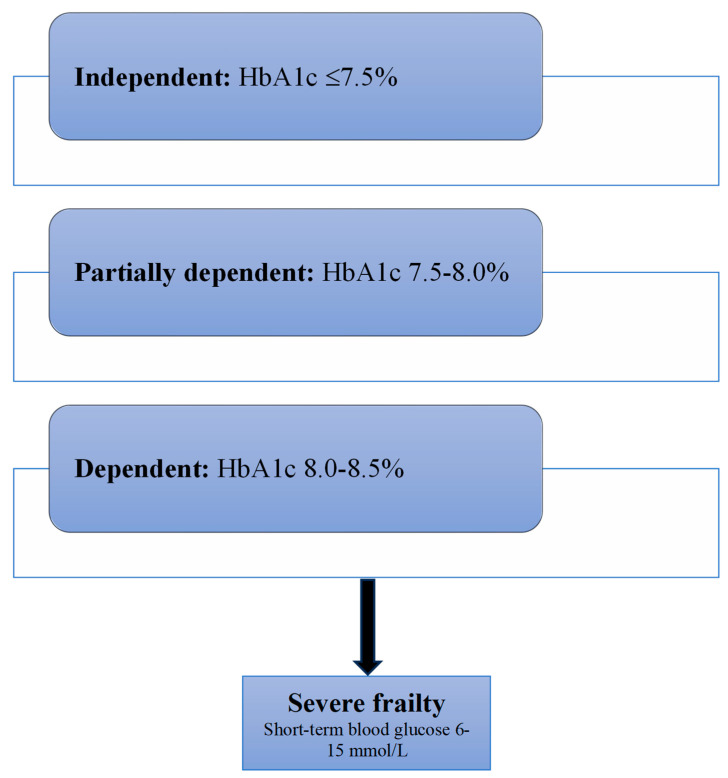
Suggested glycaemic targets in frail older people with diabetes based on function. With progression to severe frailty and short life expectancy, the focus will be on short-term rather than long-term glycaemia.

**Figure 3 diseases-13-00249-f003:**
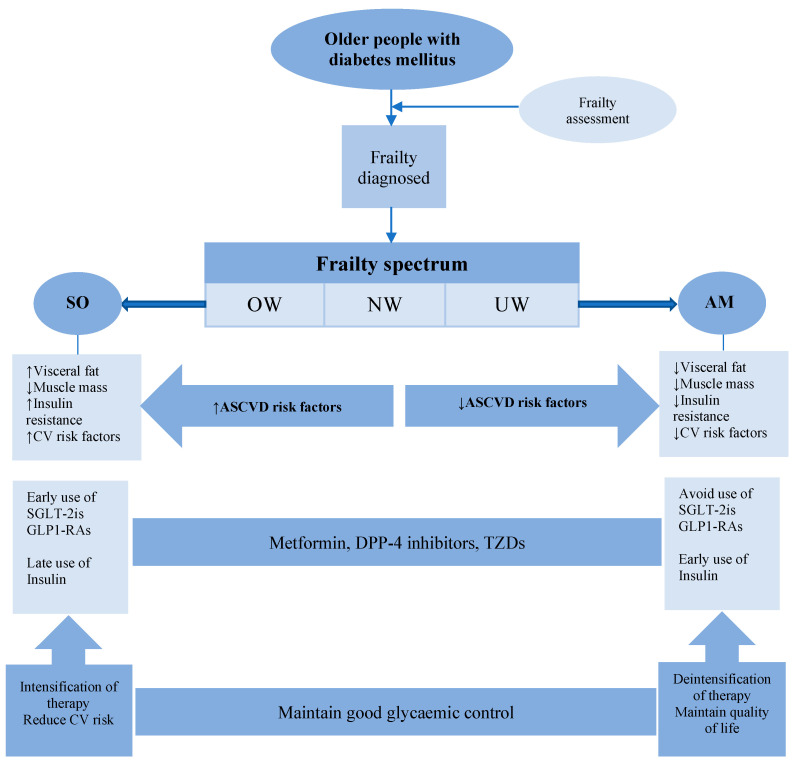
Choice of hypoglycaemic therapy in frail older people with diabetes. Clinicians should be consider hypoglycaemic agents with low threshold of hypoglycaemia. Sulfonylureas and meglitinides are better avoided due to risk of hypoglycaemia and alfa-glucosidase inhibitors due to gastrointestinal side effects. Metabolic frailty phenotype should be considered when using SGLT-2 inhibitors and GLP-1RAs, which should be used early in the SO phenotype and avoided in the AM phenotype. Low threshold for insulin use in the AM phenotype due to its anabolic properties. SO = sarcopenic obese, OW = Overweight, NW = Normal weight, UW = Underweight, AM = Anorexic malnourished, CV = Cardiovascular, ASCVD = Atherosclerotic cardiovascular disease, SGLT-2i = Sodium glucose cotransporter-2 inhibitors, GLP-1RAs = Glucagon like peptide receptor agonists, DPP-4 = Dipeptidyl peptidase, TZDS = Thiazolidinediones.

**Figure 4 diseases-13-00249-f004:**
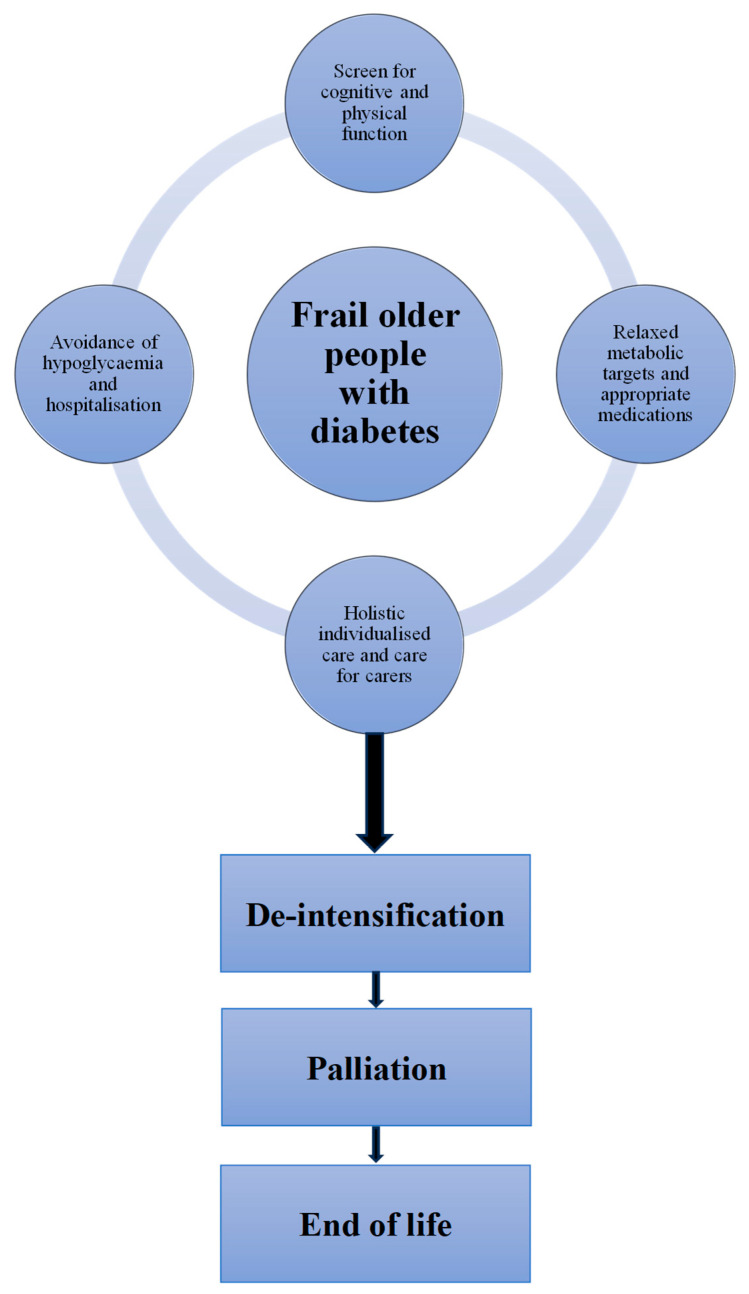
Practical considerations in the management of frail older people with diabetes.

**Table 1 diseases-13-00249-t001:** Frailty assessment tools.

Tool	Criteria	Advantage
**Fried frailty phenotype** [56]	5-point scale: Weight loss, weakness assessed by grip strength, self-reported exhaustion, reduced physical activity and slowness measured by gait speed	Identifies robust (score 0), pre-frail (score 1–2) and frail (score >3) individuals but requires two practical measurements
**Clinical frailty scale** [57]	9-point scale that describes patients’ functional characteristics and categorises them from very fit to severely frail	Uses clinical description and pictographs to stratify older people according to level of function to predict mortality or institutionalisation risk
**Frailty index** [58]	Clinical items based on data from chronic conditions, disabilities in activities of daily living, cognitive function, nutrition, visual and hearing impairments	Includes holistic data as a part of comprehensive geriatric assessment
**FRAIL scale** [59]	5-point scale: Fatigue, resistance, ambulation, illness and loss of weight	Can be self-assessed and does not require measurements by health care professionals
**Edmonton frail scale** [60]	9 domains: General health, social support, cognition, physical function, mental state, independence, nutrition, pharmacological condition and continence	Can be completed by people without special training in geriatric medicine
**Electronic frailty index** [61]	Uses the cumulative deficit model to identify and score frailty based on routine patient–general practitioner interactions	Can be used to screen for whole practice population > 65 years old
**Frailty trait scale** [62]	Evaluates three dimensions of nutrition, physical activity and nervous system	Can predict hospitalisation and mortality
**Continuous Frailty Scale** [63]	Frailty conceptualised as continuous rather than categorical construct, assessed by 5 Fried criteria	Provides risk stratification for mortality and disability beyond Fried scale

**Table 2 diseases-13-00249-t002:** Effect of glycaemic control on development of frailty.

Study	Population	Aim	Findings
**García-Esquinas E et al., prospective, Spain, 2015** [85]	346 subjects with DM, mean (SD) age 69.4 (6.4) Y, F/U 3.5 Y	Investigate mechanism of frailty in DM	↑ 1% unit in HbA1c increased risk of frailty, OR 1.48, 95% CI 1.20 to 1.81
**Zaslavsky O et al., prospective, US, 2016** [86]	200 subjects with and 1648 without DM	Explore association of blood glucose levels with frailty	U-shaped relation of HbA1c and frailty Reference HbA1c: 7.6% A. ↑ HbA1c (8.2): HR 1.3, 95% CI 1.1 to 1.6 A. ↓ HbA1c (6.9%): HR 1.4 (1.1 to 1.8)
**Morita T et al., prospective, Japan, 2017** [87]	184 patients with DM aged 65–94 Y, F/U 5Y	Examine whether low HbA1c increases risk of support/care need certification	HbA1c < 6.0% increased risk of support/care need certification: HR 3.45, 95% CI 1.02 to 11.6, *p* = 0.046 compared to HbA1c 6.5–7%
**Yanagita I et al., retrospective, Japan, 2018** [88]	132 hospitalised patients with DM, mean (SD) age 78.3 (8.0) Y	Explore risk factors of frailty including HbA1c	A. Mean (SD) HbA1c significantly lower in frail compared to non-frail patients, 7.13 (0.99) vs. 7.27 (1.04), *p* < 0.001 B. HbA1c inversely correlated with CFS (r = −0.31, *p* < 0.01) C. HbA1c independently predicted frailty (ß = −0.367, *p* < 0.01)
**Aguayo GA et al., prospective, UK, 2019** [89]	5377 subjects, median (IQR) age 70 (65, 77) Y, F/U 10 Y	Investigate if subjects with DM or high HbA1c have different frailty trajectories with ageing	A. Subjects with DM had higher frailty throughout later life B. ↑ HbA1c associated with frailty (b = 4.2, 95% CI 2.5 to 5.9)
**Hyde Z et al., cross-sectional, Australia, 2019** [90]	141 Aboriginal Australians, mean (SD) age 62.2 (11.1) Y	Investigate whether HbA1c is associated with frailty	A. 51.1% subjects had DM, 59.6% frail B. Mean (SD) HbA1c 7.9% (2.1) in subjects with DM and 6.1% (0.9) in those without C. Frailty greater with higher HbA1c (*p* = 0.025) D. HbA1c ≥ 6.5% associated with frailty (OR 2.39, 95% CI 1.17 to 4.89)
**Bilgin S et al., cross-sectional, Turkey, 2020** [91]	101 subjects with DM, 41 frail, mean (SD) age 64.2 (8.0) Y, 60 non-frail, mean (SD) age 62.2 (7.0) Y	Examine clinical and laboratory indices of frail and non-frail patients with DM	Edmonton frail score correlated positively with HbA1c (r = 0.44, *p* < 0.001)
**Mackenzie HT et al., retrospective, Canada, 2020** [92]	400 hospitalised subjects, mean (SD) age 81.4 (8.1) Y	Investigate association of frailty and DM with hospital outcomes	A. Mean admission glucose decreased with increased frailty (*p* = 0.003); mean (SD) glucose 13.0 (8.4) mmol/L for mild, 9.0 (3.4) mmol/L for moderate and 8.6 (3.4) mmol/L for severe frailty, respectively B. Nine patients had hypoglycaemia on admission, all were frail
**Fung E et al., prospective, Hong Kong, 2021** [93]	215 subjects with DM, median (IQR) age 74 (71–78) Y	Investigate association of blood glucose levels with frailty	Less glucose control (TIR ≤ 50% of time) associated with frailty as compared to better control, median FI 0.23 (IQR 0.17–0.30) vs. 0.18 (0.13–0.25), *p* = 0.0045
**Kong L et al., cross-sectional, China, 2021** [94]	291 older people, median (IQR) age 69 (67–72) Y with DM	Identify predictors of frailty in community-dwelling older people with DM	Higher HbA1c was significant predictor of frailty: OR 1.434, 95% CI 1.045 to 1.968
**Lin CL et al., cross-sectional, Taiwan, 2022** [95]	248 subjects with DM, mean (SD) age 73.9 (5.9) Y	Investigate risk factors associated with frailty.	↑ HbA1c (≥8.0%) and frequent hyperglycaemia (16.7 mmol/L) associated with frailty (*p* = 0.038 and 0.001, respectively)

DM = Diabetes mellitus, SD = Standard deviation, Y = Year, F/U = Follow-up, OR = Odds ratio, CI = Confidence interval, HR = Hazard ratio, CFS = Clinical frailty scale, IQR = interquartile range, TIR = Time in range, FI = Frailty index.

**Table 3 diseases-13-00249-t003:** Advantages and disadvantages of hypoglycaemic agents in frail older people with diabetes mellitus.

Agent	Advantage	Disadvantage
**Metformin**	Less risk of hypoglycaemia, cardiovascular benefit, weight-neutral, may have positive effect on frailty, depression and dementia but effect may be confounded by vitamin B12 deficiency	Not suitable for patients with significant weight loss or at risk of lactic acidosis, such as those with renal impairment, dehydration, heart failure and acute illness
**Thiazolidinediones**	Suitable for patients with renal impairment, less risk of hypoglycaemia, may have positive effect on frailty and dementia but no data on depression	Not suitable for patients with fluid retention or heart failure, concerns over use in patients with osteoporosis
**Sulfonylureas**	Suitable for patients with renal impairment and lean patients with less risk of hypoglycaemia	Not suitable for obese individuals or those at risk of recurrent hypoglycaemia, particularly those living alone; may have negative effects on frailty, depression and cognition; long-acting sulfonylureas should be avoided
**Meglitinides**	Short-acting, suitable for patients with erratic eating patterns	Risk of hypoglycaemia and weight gain, may have negative effects on frailty, depression and cognition
**Alpha-glucosidase inhibitors**	Less risk of weight gain and hypoglycaemia	Weak hypoglycaemic action, gastrointestinal side effects
**Dipeptidyl peptidase-4 (DPP-4) inhibitors**	Low risk of hypoglycaemia, weight loss, may have positive effect on depression, cognition and muscles	Gastrointestinal side effects, dose mostly needs to be adjusted with renal impairment
**Sodium glucose cotransporter-2 (SGLT-2) inhibitors**	Low risk of hypoglycaemia; weight loss in obese frail individuals; not enough data on effect on frailty, depression or cognition.	Not suitable for frail older people with weight loss and heavy glycosuria; increases risk of urinary tract infections, candidiasis, dehydration and hypotension.
**Glucagon like-1 receptor agonists (GLP-1RAs)**	Low risk of hypoglycaemia; weight loss in obese frail individuals; not enough data on effect on frailty, depression or cognition	Injectable, weight loss in frail anorexic individuals, not suitable in renal failure, nausea is common, possible risk of pancreatitis
**Insulin**	Effective, tailored rapidly to changes in need, improves quality of life; not enough data of effect on frailty, depression or cognition	High risk of hypoglycaemia and weight gain

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
