# Peer review of "Practical Considerations in the Management of Frail Older People with Diabetes"

_diseases, 2025, doi:10.3390/diseases13080249_

Round 1
Reviewer 1 Report
Comments and Suggestions for Authors
Review of "Practical considerations in the management of frail older people with diabetes"
This review article summarized the management of frail older people with diabetes. This reviewer thinks "Reducing Hypoglycaemia" and "Avoiding Hospitalisation" were well summarized. This reviewer has a few comments.
This reviewer understands that frail is known to involve three factors: physical, mental, and social. Therefore, this reviewer would like to see a summary of management in frail older people with diabetes for each of these three elements.
Author Response
Many thanks for your comments and suggestions to improve the manuscript.
Frailty is known to involve three factors: physical, mental, and social.
We have taken this into account, considered it in text and added a new Box 1 to summarise the screening tools for the three aspects of frailty.
We also added a new Table 3 for advantages and disadvantages of hypoglycaemic agents and considered the multidimensional aspects of frailty.
Reviewer 2 Report
Comments and Suggestions for Authors
The manuscript addresses a highly relevant and increasingly important topic in geriatric endocrinology. It offers a comprehensive overview of the intersection between frailty and diabetes in older adults and proposes practical considerations for management. However, there are several critical issues related to structure, clarity, evidentiary support, redundancy, and methodological framing that need substantial revision to improve scientific rigor and coherence.
- The manuscript often repeats the same points in different sections (e.g., the bidirectional relationship between frailty and hypoglycemia is reiterated in multiple places without new insights).
- Although the sections broadly follow a logical flow, the long subsections (especially 2.3 to 2.9) would benefit from subheadings or restructured summaries.
- The paper uses terms like "anorexic malnourished (AM) frailty phenotype" and "sarcopenic obese (SO) frailty phenotype" without adequate definitions or references to validated classification (see doi: 10.1007/s41999-025-01168-1).
- There is no clear theoretical model guiding the selection or integration of the various practical recommendations.
- The paper often mixes retrospective, cross-sectional, and anecdotal evidence without distinguishing their evidentiary weight.
- Some of the tables (e.g., Table 2 summarizing glycemic targets and frailty risk) are not critically appraised. Include commentary on limitations, sample sizes, or biases in studies cited in tables.
- Despite claiming to offer “practical considerations,” the manuscript lacks clinical pathways or decision trees. Include a clinical decision-making algorithm stratified by frailty severity and metabolic profile.
- The discussion on hypoglycemic agents lacks attention to drug-drug interactions, renal function, or cost/accessibility in real-world settings. Add a comparative table or risk-benefit matrix for different agents considering comorbidities and functional status.
- Several sentences are grammatically awkward or redundant (e.g., “...is likely to increase” is repeated extensively).
- The English needs revision at some points
Author Response
Many thanks for your comments and suggestions to improve the manuscript.
The manuscript often repeats the same points in different sections
We have reviewed this throughout the manuscript to reduce redundancies.
The long subsections (especially 2.3 to 2.9) would benefit from subheadings or restructured summaries
We have shortened sections and subsections across the manuscript.
"Anorexic malnourished (AM) frailty phenotype" and "sarcopenic obese (SO) frailty phenotype" without adequate definitions or references
We have published this new concept in detail previously and wanted to avoid duplication. However, we have added a brief definition and reference, highlighted, page 8.
There is no clear theoretical model guiding the selection or integration of the various practical recommendations
We agree, this is due to lack of specific studies in this group of patients who are largely excluded from clinical trails. We have tried to gather the available evidence as possible and also suggest practical consideration based on reasonable clinical practice as suggested overall plans should be individualised using clinician experience and patients wishes in a shared decision making manner.
The paper often mixes retrospective, cross-sectional, and anecdotal evidence without distinguishing their evidentiary weight and Some of the tables (e.g., Table 2 summarizing glycemic targets and frailty risk) are not critically appraised. Include commentary on limitations, sample sizes, or biases in studies cited in tables.
Please see above. We have added some limitations of studies investigating relationship between glycaemic control and frailty. Design of all studies is stated in Table 2. We have also highlighted that all these studies are either retrospective or cross-sectional and they are mostly reporting association rather than causation, which limit the weight of evidence provided by these studies. This is likely expected for investigation of frail older people who are difficult to recruit in trials and the evidence in these vulnerable patients remains patchy and scarce.
Despite claiming to offer “practical considerations,” the manuscript lacks clinical pathways or decision trees.
We have added a new Figure 3 to help decision making.
The discussion on hypoglycemic agents lacks attention to drug-drug interactions, renal function, or cost/accessibility in real-world settings.
We have added a new Table 3 summarising hypoglycaemic therapy.
Several sentences are grammatically awkward or redundant and the English needs revision at some points
We have reviewed sentences and grammar throughout the manuscript.
Reviewer 3 Report
Comments and Suggestions for Authors
This is an interesting and necessary manuscript, which draws attention, on the one hand, to the need to consider frailty in patients with diabetes, and on the other hand, to the necessity of individualizing the approach in relation to the progression of the disease.
I only have minor editorial comments regarding the figures: Figure 1 needs revision (there are hyphenated words in the upper part), and Figure 3 is split across two pages, which makes it difficult to read in its current form.
Author Response
Many thanks for your comments and suggestions to improve the manuscript.
Figure 1 needs revision (there are hyphenated words in the upper part), and Figure 3 is split across two pages, which makes it difficult to read in its current form.
Thank you for spotting this. We have corrected it and will be submitting an image version of figures to avoid it happening again.
Round 2
Reviewer 2 Report
Comments and Suggestions for Authors
The manuscript can be now accepted for publication.
Author Response
Thank you. The English has been reviewed throughout the manuscript.